# PROTOCODE: PROTOTYPE-DRIVEN INTERPRETABILITY FOR CODE GENERATION IN LLMS

## ABSTRACT

Since the introduction of Large Language Models (LLMs), they have been widely adopted for various tasks such as text summarization, question answering, speech-to-text translation, and more. In recent times, the use of LLMs for code generation has gained significant attention, with tools such as Cursor and Windsurf demonstrating the ability to analyze massive code repositories and recommend relevant changes. Big tech companies have also acknowledged the growing reliance on LLMs for code generation within their codebases. Although these advances significantly improve developer productivity, increasing reliance on automated code generation can proportionally increase the risk of suboptimal solutions and insecure code. Our work focuses on automatically sampling In-Context Learning (ICL) demonstrations which can improve model performance and enhance the interpretability of the generated code. Using AST-based analysis on outputs from the MBPP test set, we identify regions of code most influenced by the chosen demonstrations. In our experiments, we show that high-quality ICL demonstrations not only make outputs easier to interpret but also yield a positive performance improvement on the pass@10 metric. Conversely, poorly chosen ICL demonstrations affected the LLM performance on the pass@10 metric negatively compared to the base model. Overall, our approach highlights the importance of efficient sampling strategies for ICL, which can affect the performance of the model on any given task.

## 1 INTRODUCTION

In recent years, Large Language Models (LLMs) have gained significant traction in the fields of code completion and code filling. This growth has been fueled by the availability of large-scale open-source datasets such as The vault Manh et al. (2023), CodeSearchNet Husain et al. (2020), CodeXGlue Lu et al. (2021) and many others. Alongside these datasets, we have also witnessed the emergence of open-source models designed specifically for code-related tasks, including the CodeLlama series Rozière et al. (2024), Qwen Coder Hui et al. (2024) series, and StarCoder series Li et al. (2023a). In parallel, closed-source models such as GPT-4o OpenAI et al. (2024) and Claude Code Anthropic (2025) have been widely adopted by various big tech companies for generating production-ready code. Despite these advancements, most of these models remain difficult to interpret in the context of code generation. While a variety of interpretability methods have been developed to interpret the outputs generated by LLMs and foster trust in their usage across domains, many of these approaches are generic and not specifically tailored for code generation tasks. Some methods, however, are focused on interpretability in code generation. For instance, Code-Q Palacio et al. (2025) identifies influential tokens that guide the model's output, but it requires repeated sampling and generation, which introduces significant computational overhead during inference.

Another method, ASTrust Palacio et al. (2024), leverages Abstract Syntax Trees (ASTs) by using model-generated token probabilities. Tokens are mapped to code level subsets, which are then grouped into terminal and non-terminal nodes within the AST. Each non-terminal node is represented by the aggregated confidence of its associated terminal nodes. However, this approach requires storing the probability distribution over the entire vocabulary at every step of generation, which scales poorly as the output length increases. To address these challenges, we propose a manifold-based sampling strategy that automatically samples a set of ICL demonstrations from a given dataset. These demonstrations enable interpretability by combining attribution and AST-based

analysis. Our method segments the generated code into interpretable regions, such as Iterations, Data structures, etc., allowing users to understand which regions of the generated code are most affected by the sampled demonstrations. To the best of our knowledge, we are the first to unify prototype-driven ICL sampling with AST-grounded attribution for code interpretability.

- **Prototype Sampling via Joint Manifold and Metric Learning:** Our method introduces a principled approach to sample In-Context Learning (ICL) demonstrations by combining *piecewise-linear manifold learning* and *proxy anchor–based metric learning*. This joint formulation ensures that the sampled prototypes are not only *geometrically faithful*—capturing the local data structure—but also *semantically discriminative*.

- **Prototype-Gradient Attribution for AST-Grounded Interpretability:** We propose a novel attribution mechanism using the gradient of similarity between prototype and token embeddings to estimate token-level influence. These scores are then propagated through the *Abstract Syntax Tree (AST)* to produce *faithful, syntax-aware confidence maps*, enabling both *local (node-level)* and *global (category-level)* interpretability of generated code—while avoiding the memory overhead of storing token probabilities.

## 2 RELATED WORK

According to Bilal et al. (2025), explainability techniques in AI systems can be broadly divided into three categories: (1) post hoc explanations, (2) intrinsic interpretability, and (3) human-centered explanations. Post hoc explanation methods aim to interpret a model's decisions after predictions have been made. Common approaches include Local Interpretable Model-Agnostic Explanations (LIME) Ribeiro et al. (2016), Shapley Additive Explanations (SHAP) Lundberg & Lee (2017). LIME provides local explanations by identifying the most important features for a single prediction. Similarly, SHAP evaluates the contribution of each feature by measuring changes in the prediction when features are systematically removed. In addition, gradient-based methods such as SmoothGrad Smilkov et al. (2017) and Integrated Gradients Sundararajan et al. (2017) calculate model gradients with respect to input features to determine the sensitivity of the model's output to each feature.

Intrinsic interpretability, in contrast, focuses on designing model architectures so that their behavior is inherently explainable. One example is concept bottleneck models Koh et al. (2020), which were extended to large language models (LLMs) by Sun et al. (2025) for sentence classification task. Their approach generates concepts for each class, making the classification process directly interpretable. However, this approach faces limitations in generating suitable concepts for diverse tasks and does not scale well to text generation. Another related method, Proto-lm Xie et al. (2023) , extends prototype networks to text classification. Instead of generating concepts like concept bottlenecks, it learns trainable prototypes and maps them to the nearest training samples for interpretability.

A particularly influential method within intrinsic interpretability is Chain-of-Thought (CoT) Wei et al. (2023), which generates intermediate reasoning steps. CoT has been shown to improve both plausibility and task performance compared to demonstrations that provide only the final answers Wei et al. (2023) Cobbe et al. (2021). Building upon this, Self-Consistency Wang et al. (2023) was proposed as an extension of CoT. This method prompts the model to produce multiple reasoning chains and answers, and then selects the final output using a majority vote across the answers. Although effective, Self-Consistency only ensures correctness of the final prediction, without verifying whether the reasoning chains themselves are valid or faithful. To address this, SEA-CoT Wei Jie et al. (2024) was introduced. SEA-CoT evaluates generated reasoning chains based on the implication with the task context and the overlap of the token level, ensuring that both the reasoning process and the final answer align more closely with the task requirements. However, as stated by Jacovi & Goldberg (2020), the reasoning chains from LLM often appear plausible to humans but are not necessarily faithful to the true decision-making process of the LLM. Plausibility refers to how convincing the interpretation is to humans, while faithfulness measures the degree to which it truly represents the internal reasoning of the LLM.

Most of the above methods are designed for generic tasks, with a limited focus on code-specific applications. The method ASTrust was developed specifically for interpretability in code generation. It builds Abstract Syntax Trees (ASTs) to align with program structure and assigns confidence scores to non-terminal nodes by aggregating probabilities from their terminal nodes. These scores are de-

rived from token-level probabilities output by the model. Ma et al. (2024) demonstrates that LLMs already possess strong syntactic awareness, rivaling AST-based static code analysis. However, the method ASTrust has key limitations: its token sampling method is not well justified. Greedy sampling ignores the advantages of stochastic approaches, while stochastic sampling requires storing probabilities for all vocabulary tokens at every step an impractical, memory-intensive process. In contrast, our method avoids this heavy storage by relying on attribution-based prototype influence, which captures the effect of sampled demonstrations without requiring full vocabulary distributions. As a result, our approach preserves the benefits of stochastic sampling Shi et al. (2024) while remaining significantly more scalable and practical for code generation interpretability.

## 3 METHODOLOGY

Prototype-based approaches provide an interpretable mechanism to associate each class with representative examples, commonly referred to as prototypes. A simple baseline is to define prototypes using statistics such as class means or medoids in the embedding space. However, these statistical summaries fail to capture the intrinsic geometry of the representation space: they are vulnerable to outliers, insensitive to intra-class multimodality, and often yield prototypes that are statistically central yet semantically uninformative.

To overcome these shortcomings, we turn to the manifold perspective. The manifold hypothesis Cayton (2005) posits that high-dimensional representations lie on low-dimensional manifolds. Leveraging this structure allows prototypes to be sampled from regions that faithfully capture the local geometry of the data, rather than from globally averaged or distorted positions in embedding space. While classical manifold learning techniques such as t-SNE van der Maaten & Hinton (2008), UMAP McInnes et al. (2020), and LLE Roweis & Saul (2000) emphasize neighborhood preservation, they often distort local dependencies or fail to maintain global structure. We therefore adopt a piecewise-linear manifold learning strategy, which decomposes nonlinear manifolds into locally linear regions.

While geometry preserves structural fidelity, it does not guarantee that prototypes are discriminative across classes. To enforce both intra-class compactness and inter-class separation, we integrate metric learning objectives. Traditional formulations such as triplet or contrastive loss require pre-specified prototypes and extensive mining, making them inefficient and unstable. Instead, we employ Proxy-Anchor loss, which introduces learnable class-level proxy vectors to directly optimize intra-class cohesion and inter-class margins. After training, each learned proxy vector is mapped to its nearest training instance using euclidean distance.

As highlighted in Rodriguez-Cardenas et al. (2023), in the context of ICL, the selection of demonstrations plays a crucial role in model performance. In our approach, we dedicate considerable effort to identifying the most suitable prototypes (ICL examples) for each LLM. Our method can be divided into two main components. In the first stage, we initialize a simple neural network $h_\theta$ and train it on Dataset $D$ to jointly optimize manifold learning and metric learning objectives. Once the training is complete, the learned proxy vectors are employed to sample prototypes.

### 3.1 DATASET

We have used the Magicoder-OSS-Instruct-75K Wei et al. (2024) for sampling the prototypes. This dataset consists of 75,000 synthetic instruction-following examples generated using OSS-Instruct; it contains 9 programming languages. For every query, it has a programming language id, the query, and the code solution. For every sample in the dataset, we have used the following prompt structure to format all the samples in the dataset.

Prompt Structure: *"This is the query being assigned:"+" "+ [/Q]+" "+"The following is the code solution to the query"+" "+[/S]"*. Where the placeholders *[/Q]* and *[/S]* are for query and code solution respectively. After formatting the prompts, we use the respective Large Language model($M$) to encode the final prompts into the latent representations ($z$). We simultaneously label encode the programming language ID for using them as class labels; this method gives us 9 different classes, and for each sample in the dataset, we will be storing the encoded label ($l$) and the latent representation ($z$) as pairs in dataset $D$.

## 3.2 TRAINING OVERVIEW

As mentioned in 3, our method consists of two stages. In the first stage of our method, we initialize a simple neural network $h_\theta$ Tab 3 and train it on Dataset $D$ to jointly optimize manifold learning 3 and metric learning objectives 1. The neural network $h_\theta$ learns to map the high-dimensional encoded representations into lower dimensions. Before the training process, we initialize the proxies $\theta_q$ and $\theta_m$. Here both the proxies are unique for each class and initialized randomly with $\theta_q = \theta_m$. The proxy vector $\theta_q$ is updated via back-propagation, and the proxy vector $\theta_m$ is updated via the Momentum update He et al. (2020) where $\gamma$ is the momentum constant, $[\theta_k \leftarrow \gamma\theta_k + (1-\gamma)\theta_q]$

During training, for every mini-batch $B$ we build linear piecewise manifolds as outlined in 3.3. For every point in $B$, we then compute the manifold-based similarity following the procedure in 3.4. This similarity measure is used to compute the manifold point-to-point loss $\mathcal{L}_{\text{manifold}}$. At the same time, we compute the Proxy Anchor loss $\mathcal{L}_{\text{PA}}$ using randomly initialized class proxies $\theta_q$ and latent representations $z$ in batch $B$. The final loss is computed as, $\mathcal{L}_{\text{total}} = \mathcal{L}_{\text{PA}} + \mathcal{L}_{\text{manifold}}$.

While the manifold loss preserves local geometric structure, the Proxy-Anchor loss promotes intra-class compactness and inter-class separation, thereby facilitating the discriminative learning of prototypes. Across epochs, the network parameters are updated via backpropagation. After training, the momentum-updated proxies $\theta_m$ are used to select the nearest training instance as the prototype for each class, yielding a single prototype per class for the subsequent stage.

After the completion of the training process, we proceed to the second stage, where we generate the code completions using the prototypes. After that, we utilize the encoded latent representations of the prototypes to calculate the confidence score per token for AST analysis. For each code completion, we compute a prototype attribution-based score to quantify the influence of the prototypes on the generated code. Specifically, influence is measured by the attribution between the sampled demonstrations and the code completions. Finally, we perform an AST analysis to analyze how the prototypes impact the syntactic structure of the generated code.

## 3.3 MANIFOLD CONSTRUCTION

Based on the Manifold hypothesis, we can assume that the encoded latent representations $z$, which are inherently complex and non-linear, can be locally approximated into smaller chunks of linear regions. Our approach leverages this structural assumption to automatically identify representative prototypes that capture the essential characteristics of each action class.

To efficiently approximate the structure of the linear data manifolds, we adopt a piecewise linear manifold learning method which constructs localized $m$-dimensional linear submanifolds around selected anchor points. Given a mini batch $B$ containing $N$ data points, we randomly select $n$ of them to serve as anchors. For each anchor point $h_\theta(z_i)$, we initially collect its $m-1$ nearest neighbors in the encoded representation space based on Euclidean distance to form the neighborhood set $X_i$.

The manifold expansion process proceeds iteratively by attempting to add the $m$-th nearest neighbor to $X_i$. After each addition, we recompute the best-fit $m$-dimensional submanifold using PCA and assess whether all points in $X_i$ can be reconstructed with a quality above a threshold $T\%$. If the reconstruction quality remains acceptable, the new point is retained in $X_i$; otherwise, it is excluded. This evaluation is repeated for subsequent neighbors $N(h_\theta(x_i))_j$ for $j \in \{m+1, \ldots, k\}$, gradually constructing a local linear approximation of the manifold.

The final set $X_i$ comprises all points in the anchor's neighborhood that lie well within an $m$-dimensional linear submanifold. A basis for this submanifold is computed by applying PCA to $X_i$ and extracting the top $m$ eigenvectors. We choose PCA for this task as it can effectively construct the lower-dimensional manifolds for the locally linear regions.

## 3.4 TRAINING OBJECTIVES

**Proxy Anchor Loss:** We use a modified version of proxy anchor loss with Euclidean distance instead of cosine similarity:

$$\mathcal{L}_{\text{PA}} = \frac{1}{|\Theta_+|} \sum_{\theta_q \in \Theta_+} \log \left( 1 + \sum_{z \in \mathcal{Z}_{\theta_q}^+} \exp\left(-\alpha \cdot \left(\|h_\theta(z) - \theta_q\|_2 - \epsilon\right)\right) \right) \quad (1)$$

$$+ \frac{1}{|\Theta|} \sum_{\theta_q \in \Theta} \log \left( 1 + \sum_{z \in \mathcal{Z}_{\theta_q}^-} \exp\left(\alpha \cdot \left(\|h_\theta(z) - \theta_q\|_2 - \epsilon\right)\right) \right) \quad (2)$$

Here, $\Theta$ denotes the set of all proxies, where each proxy $\theta_q \in \Theta$ serves as a representative vector for a class. The subset $\Theta_+ \subseteq \Theta$ includes only those proxies that have at least one positive embedding in the current batch $B$. For a given proxy $\theta_q$, the latent representations $\mathcal{Z}$ in $B$ (where $z \in \mathcal{Z}$) are partitioned into two sets: $\mathcal{Z}_{\theta_q}^+$, the positive embeddings belonging to the same class as $\theta_q$, and $\mathcal{Z}_{\theta_q}^- = \mathcal{Z} \setminus \mathcal{Z}_{\theta_q}^+$, the negative embeddings. The scaling factor $\alpha$ controls the sharpness of optimization by amplifying hard examples when large (focusing gradients on difficult pairs) or smoothing training when small (spreading weight across all pairs). The margin $\epsilon$ enforces a buffer zone between positives and negatives by requiring positives to be closer to their proxies and negatives to be sufficiently farther away.

**Manifold Point-to-Point Loss:**  This loss helps in estimating the point to point similarities preserving the geometric structure:

$$\mathcal{L}_{\text{manifold}} = \sum_{i,j} \left( \delta \cdot (1 - s(z_i, z_j)) - \|h_\theta(z_i) - h_\theta(z_j)\|_2 \right)^2 \quad (3)$$

where $s(z_i, z_j)$ is the manifold similarity computed as:

$$s(z_i, z_j) = \frac{s'(z_i, z_j) + s'(z_j, z_i)}{2}$$

with $s'(z_i, z_j) = \alpha(z_i, z_j) \cdot \beta(z_i, z_j)$, where:

$$\alpha(z_i, z_j) = \frac{1}{(1 + o(z_i, z_j)^2)^{N_\alpha}}$$

$$\beta(z_i, z_j) = \frac{1}{(1 + p(z_i, z_j))^{N_\beta}}$$

In equation 3, $h_\theta$ is a simple neural network with a structure specified in Table 3 and $\delta$ is a scaling factor that determines the maximum separation between dissimilar points. The loss encourages Euclidean distances in the embedding space to match manifold-based dissimilarities $1 - s(z_i, z_j)$, ensuring that the learned metric space respects the underlying manifold structure. $o(z_i, z_j)$ is the orthogonal distance from point $z_i$ to the manifold of point $z_j$, and $p(z_i, z_j)$ is the projected distance between point $z_j$ and the projection of $z_i$ on the manifold. The parameters $N_\alpha$ and $N_\beta$ control how rapidly similarity decays with distance, with $N_\alpha > N_\beta$ ensuring that similarity decreases more rapidly for points lying off the manifold than for points on the same manifold.

**Distance Calculation.**  For each point pair $(z_i, z_j)$, the distances $o(z_i, z_j)$ and $p(z_i, z_j)$ are calculated using the manifold basis vectors $P_j$ associated with point $z_j$. The projection of $z_i$ onto $P_j$ is computed as $\text{proj}_{P_j}(z_i) = z_j + \sum_k \langle z_i - z_j, v_k \rangle v_k$, where $v_k$ are the basis vectors of $P_j$. The orthogonal distance is then $o(z_i, z_j) = \|z_i - \text{proj}_{P_j}(z_i)\|_2$, and the projected distance is $p(z_i, z_j) = \|\text{proj}_{P_j}(z_i) - z_j\|_2$. This process is repeated for all point pairs, capturing the full geometric structure of the data manifold.

## 4  RESULTS

We evaluated the effectiveness of different sampling methods by applying them as in-context learning (ICL) examples on the MBPP test set Austin et al. (2021). To demonstrate the effectiveness of

Table 1: Performance comparison across different models and methods on the MBPP dataset

| Model →
Method ↓ | Qwen3-0.6B | | Llama3.2-1B | | Falcon3-1B | | Starcoder-1B-base | | Qwen2.5coder-0.5B | | Codellama-7B | |
|---|---|---|---|---|---|---|---|---|---|---|---|---|
| | $pass@1$ | $pass@10$ | $pass@1$ | $pass@10$ | $pass@1$ | $pass@10$ | $pass@1$ | $pass@10$ | $pass@1$ | $pass@10$ | $pass@1$ | $pass@10$ |
| base | 0.011 | 0.048 | 0.007 | 0.042 | 0.010 | 0.063 | 0.008 | 0.040 | 0.041 | 0.116 | 0.021 | 0.116 |
| diversity | 0.0076 | 0.037 | 0.012 | **0.061** | 0.010 | 0.042 | 0.002 | 0.011 | 0.021 | 0.063 | 0.007 | 0.035 |
| similarity | 0.009 | 0.050 | **0.013** | 0.050 | 0.011 | 0.050 | 0.007 | 0.032 | 0.023 | 0.069 | 0.018 | 0.079 |
| mbpp | 0.006 | 0.024 | 0.007 | 0.042 | 0.002 | 0.018 | 0.005 | 0.004 | 0.021 | 0.095 | 0.009 | 0.039 |
| prototypes(ours) | **0.019** | **0.059** | 0.010 | 0.058 | **0.020** | **0.068** | **0.012** | **0.050** | **0.048** | **0.122** | **0.030** | **0.122** |

Table 2: Performance comparison across different models and methods on the MBPP+ Dataset

| Model →
Method ↓ | Qwen3-0.6B | | Llama3.2-1B | | Falcon3-1B | | Starcoder-1B-base | | Qwen2.5coder-0.5B | | Codellama-7B | |
|---|---|---|---|---|---|---|---|---|---|---|---|---|
| | $pass@1$ | $pass@10$ | $pass@1$ | $pass@10$ | $pass@1$ | $pass@10$ | $pass@1$ | $pass@10$ | $pass@1$ | $pass@10$ | $pass@1$ | $pass@10$ |
| base | 0.0078 | 0.0396 | 0.005 | 0.037 | 0.009 | **0.058** | 0.008 | 0.037 | 0.031 | 0.100 | 0.015 | 0.090 |
| diversity | 0.0067 | 0.031 | 0.007 | 0.045 | 0.007 | 0.034 | 0.002 | 0.011 | 0.017 | 0.048 | 0.006 | 0.026 |
| similarity | 0.0061 | 0.037 | **0.008** | **0.054** | 0.007 | 0.042 | 0.006 | 0.029 | 0.017 | 0.055 | 0.013 | 0.067 |
| mbpp | 0.002 | 0.016 | 0.005 | 0.042 | 0.001 | 0.013 | 0.004 | 0.032 | 0.016 | 0.077 | 0.006 | 0.040 |
| prototypes(ours) | **0.016** | **0.050** | 0.007 | 0.050 | **0.015** | 0.050 | **0.010** | **0.046** | **0.039** | **0.108** | **0.024** | **0.103** |

our method we have used 2 sets of models for experimentation, the first set consisting of generic models of Qwen3 Yang et al. (2025), Llama-3.2 AI (2024),Falcon-3 Team (2024) and for the second set we have used code heavy pre-trained models Starcoder-base Li et al. (2023b), Qwen2.5-Coder Hui et al. (2024), Codellama Rozière et al. (2024).

The results are reported on a scale from $[0, 1]$, where 0 is the lowest and 1 is the highest (For instance, 0.1 can be interpreted as 10%). While the numerical margins may appear small at first glance, even modest gains in code completion represent substantial improvements. For context, GPT-4-1106 ope (2023), which is estimated to be at least $1000\times$ larger than the models used for our experiments, achieves a score of $0.786$ on the MBPP test set. This comparison highlights an important distinction: in many benchmarks, partial overlap between a generated solution and the reference solution may yield a nonzero score even if the final answer is incorrect. In contrast, code benchmarks are more stringent, as each generated program is independently evaluated against unit test cases. Therefore, even incremental improvements in $Pass@k$ metrics are highly significant for code generation tasks.

The Qwen2.5-coder model, despite having fewer parameters than Codellama, achieves comparable performance on both the $Pass@1$ and $Pass@10$ metrics across the MBPP and MBPP+ test sets. Among all comparisons, the similarity-based sampling method surpasses our approach only for the Llama3.2 model; in every other case, our method consistently outperforms alternative strategies across all models. As noted by Rodriguez-Cardenas et al. (2023), within the ICL setting, the quality of selected demonstrations can also negatively affect model performance.

For the Qwen3 and Qwen2.5-coder models, using demonstrations sampled from methods other than the prototype-based approach leads to a decline in performance on both MBPP and MBPP+. A similar trend is observed for the Starcoder and Codellama models. These results suggest that the Qwen family of models, as well as code-pretrained models in general, are particularly sensitive to the choice of ICL demonstrations. An unsuitable set of demonstrations can reduce performance compared to the base model, underscoring the importance of effective sampling strategies for ICL.

## 5 AST ANALYSIS

We perform an Abstract Syntax Tree (AST) analysis to identify which syntactic regions of the generated code are most influenced by the sampled prototypes. In the ASTrust framework, the authors employ token-level probabilities produced by the model $M$ as the confidence scores in the token set. For a sequence of tokens $w_1, w_2, \ldots, w_i$, the probability of generating the next token $w_{i+1}$ is defined as equation 4 where $M$ denotes the Large Language Model and $M(w_{1:i})$ represents the non-normalized log probabilities output by the model for the given context.

$$P(w_{i+1} \mid w_{1:i}) = \text{Softmax}(M(w_{1:i})), \tag{4}$$

Figure 1: Conceptual working of AST analysis

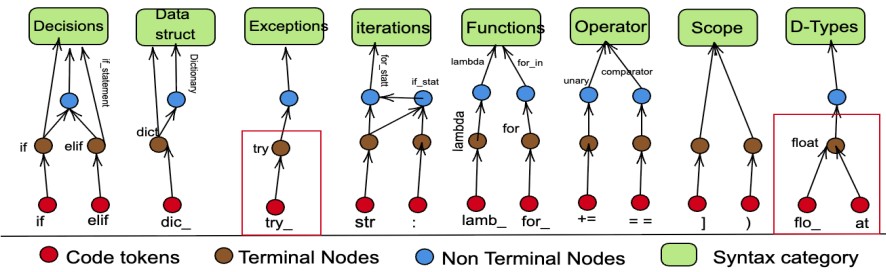

As discussed in Section 2 2, this method suffers from high memory overhead when combined with stochastic sampling strategies. To mitigate this limitation, we instead leverage attribution-based scores between the sampled prototypes and the generated code samples, and use these scores as token-level confidence in AST analysis. Concretely, for a model-generated code snippet $C$, we extract the tokens $w_i$ along with their latent representations $z_{w_i}$. Let the latent representation of a sampled prototype $p$ be $z_p$. We compute the mean prototype vector $z_a$ as $z_a = \sum_{i \in P} z_i$. Next, we compute the dot product between $z_a$ and each $z_{w_i}$, and compute its gradient with respect to $z_{w_i}$. The normalized gradients $\nabla_{z_{w_i}}$ ( equation 5) are then used as confidence scores per token in the AST analysis.

$$\nabla_{z_{w_i}} = \frac{d(z_a \cdot z_{w_i})}{dz_{w_i}} \tag{5}$$

### 5.1 SYNTAX GROUNDED EXPLANATIONS

Figure 2: Alignment & Clustering Interactions. The $\delta$ function aligns tokens $w_i$ to terminal nodes $\lambda$. Terminal and Non-terminal nodes $\lambda, \alpha \in \upsilon$ are clustered by Syntax Categories $\Lambda$

AST analysis involves using the prototype-based attribution scores as token confidence scores explained in 5. We then compute the average confidence over tokens corresponding to each AST node, and report these averages as performance values grouped by manually defined syntax categories. The process follows three steps, illustrated in Fig.1 1. In Step1, for every generated code snippet, the tokenizer splits the code into tokens $w_i$ (forming the token set $\tau$ D.1), and the model assigns a confidence score to each token as described in 5. In Step2, the token-level predictions are aligned with the respective Abstract Syntax Tree (AST) terminal nodes. Terminal nodes retain the raw confidences, whereas non-terminal nodes hierarchically store aggregated values. Together, terminal and non-terminal nodes form the subcategory set $\upsilon$ D.1. For instance, the token 'if_' from the

token set aligns with a terminal AST node but is grouped under the non-terminal node 'if_statement'. Finally, in Step 3, the analysis introduces eight syntax categories to summarize model predictions. These categories aggregate subcategories into broader, human-interpretable groups. The Syntax Categories form a fixed Category Set $\Lambda$ D.1, providing more intuitive elements for interpretation.

For example, the sub-categories 'if_statement' and 'if' are grouped under the syntax category 'Decisions' 2. Ultimately, ASTrust outputs an averaged score for each category to provide global explanations, along with an AST tree visualization that embeds confidence scores at every node for local explanations. In essence, we argue that syntax elements encode semantic information that contextualizes token-level confidence scores, though this semantic value differs depending on the granularity of the elements. For instance, tokens alone convey less interpretable meaning compared to higher-level categories. AST analysis thus serves as a post-hoc explanation framework at both local and global levels. Local explanations focus on breaking down a single code snippet into AST elements to interpret its generation, while global explanations rely on multiple generated snippets to provide a holistic view of the model through Syntax Categories (SCs) D.1.

## 5.2 Code Syntactic analysis

To assess the attribution-based confidence score of each Syntax Category (SC) for the 6 LLMs, we present an AST analysis. 3 illustrates the AST interpretability performance segregated by Syntax Categories (SCs) for each model type. The Qwen2.5 Coder and Qwen3 3 (a) models exhibit highly consistent confidence across all syntax categories, with nearly identical values. Both models demonstrate their strongest performance in Scope, Data Structures, and Functions, indicating reliability in handling structured data, variable and function scoping, and modular code organization. Moderate confidence is observed for Iteration, Decisions, Operators, and Data Types, while the lowest confidence is consistently assigned to Exception handling, suggesting potential limitations in generating or reasoning about robust error-handling constructs. Overall, these results suggest that both Qwen2.5 Coder and Qwen3 are best suited for structured programming tasks, while being less dependable for control-flow–intensive or exception-heavy code generation.

The Llama models 3 (b) exhibit broadly similar confidence trends across syntax categories, with CodeLlama consistently showing a slight advantage over Llama-3.2. Both models demonstrate their highest reliability in Data Structures, Functions, and Iteration, suggesting strong capabilities in tasks that require structured data handling, modular code organization, and loop-based constructs. Moderate confidence is observed in Scope, Decisions, Operators, and Data Types, indicating stable but less pronounced strengths. In contrast, Exception handling remains the weakest category for both models, highlighting a shared limitation in generating or reasoning about robust error-handling logic. Collectively, these results suggest that while the Llama models are well-suited for structured programming tasks, they are less dependable for exception-heavy scenarios.

The Falcon and StarCoder models 3 (c) display distinct differences in their syntax-grounded confidence. StarCoder consistently achieves higher confidence across nearly all categories compared to Falcon, indicating stronger overall reliability. Both models perform best in Scope, Data Structures, and Functions, suggesting robustness in structured programming tasks and modular code organization. StarCoder further extends this strength to Iteration and Decisions, where it shows clear improvements over Falcon, highlighting its ability to handle control flow more effectively. In contrast, Exception handling remains the weakest category for both models, underscoring a shared limitation in generating robust error-handling constructs. Taken together, these results indicate that while Falcon is moderately capable across most categories, StarCoder offers broader syntactic reliability and is better suited for tasks requiring control flow and structured data handling.

## 6 Future Works

In our experiments, prototypes were sampled exclusively from the Magicoder dataset. While this choice provided a consistent basis for evaluation, extending the analysis to additional datasets could offer a broader understanding of prototype quality. In fact, our method can naturally be applied as a global metric for ranking datasets with respect to their ability to yield effective prototypes. Another limitation arises from differences in model stability. For example, Llama3.2 5 exhibited high sensitivity to changes in nearly all hyperparameters, which led to inconsistent results on the $Pass@k$

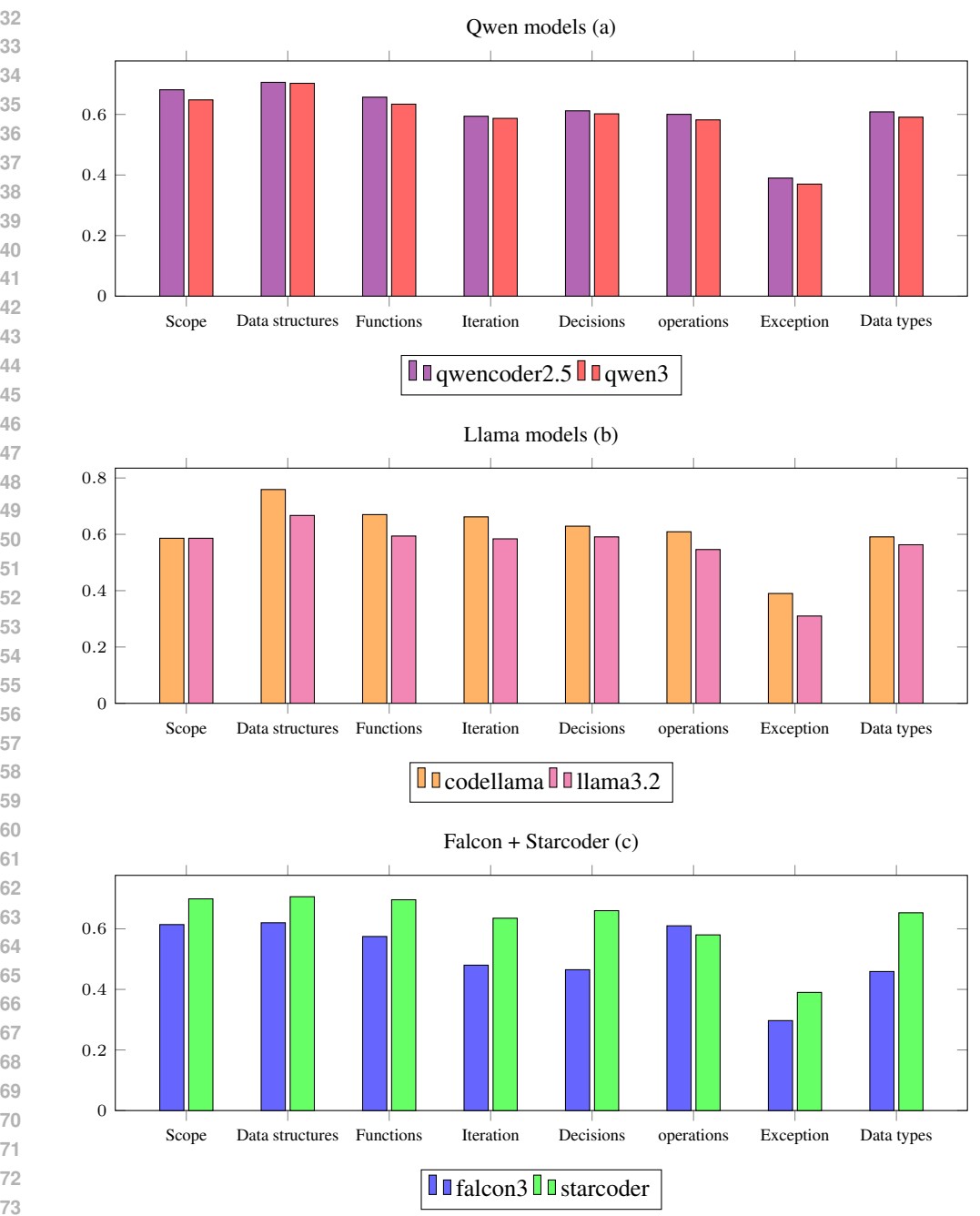

Figure 3: AST analysis on 6 LLMs

metric. In contrast, the Qwen2.5 Coder model 5 displayed only marginal sensitivity, with the exception of the $\alpha$ parameter, resulting in more stable and reliable performance. Finally, while our current approach uses sampled prototypes as in-context learning demonstrations, the framework can be extended toward pre-hoc interpretability by design. In particular, prototype steering could be explored as a mechanism for influencing model behavior, offering new avenues for both interpretability and controllability in LLMs.

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

# A    RELATED WORKS

## A.1    MANIFOLD LEARNING

The manifold hypothesis is a well-established principle in Machine Learning, which suggests that Cayton (2005):

> *Although data points often appear to have very high dimensionality, with thousands of observed features, they can typically be represented by a much smaller set of underlying parameters. In essence, the data resides on a low-dimensional manifold embedded within a high-dimensional space.*

Based on the Manifold hypothesis Manifold learning focuses on uncovering low-dimensional structures in high dimensional data. Manifold learning techniques like TSNE van der Maaten & Hinton (2008),UMAP McInnes et al. (2020), LLE Roweis & Saul (2000) and Isomap Tenenbaum et al. (2000) utilize information derived from the linearized neighborhoods of points to construct low dimensional projections of non-linear manifolds in high dimensional data.

The method Piecewise-Linear Manifolds for Deep Metric Learning Bhatnagar & Ahuja (2024) aims to train a neural network to learn a semantic feature space where similar items are close together and dissimilar items are far apart, in an unsupervised manner. This method is based on using linearized neighborhoods of points to construct a piecewise linear manifold, which helps estimate a continuous-valued similarity between data points.

## A.2    METRIC LEARNING

Metric learning aims to learn an embedding space where semantically similar samples are close and dissimilar ones are far apart. Common loss functions include **Contrastive loss** Hadsell et al. (2006),aims at making representations of positive pairs closer to each other, while pushing negative pairs further away than a positive margin. It is commonly used in tasks such as face verification or representation learning with Siamese networks. Here $(z_i, z_i')$ are embeddings of a pair, $y_i \in \{0, 1\}$ indicates similarity, and $m$ is the margin.

$$\mathcal{L} = \frac{1}{N} \sum_{i=1}^{N} \left[ y_i \, \|z_i - z_i'\|_2^2 + (1 - y_i) \, \max\left(0, m - \|z_i - z_i'\|_2\right)^2 \right]$$

**Triplet loss** Schroff et al. (2015) is another metric learning objective that enforces relative similarity by ensuring that an anchor $x_a$ is closer to a positive sample $x_p$ (same class) than to a negative sample $x_n$ (different class) by at least a margin. Unlike contrastive loss, which only considers pairwise distances, triplet loss leverages relative comparisons, making it more effective in learning discriminative embeddings for tasks such as face recognition and image retrieval, here $f(\cdot)$ is the embedding function, $m$ is the margin, $x_a$ is the anchor, $x_p$ is a positive sample, and $x_n$ is a negative sample.

$$\mathcal{L} = \frac{1}{N} \sum_{i=1}^{N} \max\left(0, \, \|f(x_a^i) - f(x_p^i)\|_2^2 - \|f(x_a^i) - f(x_n^i)\|_2^2 + m\right)$$

**Multi-class N-pair loss** Sohn (2016) generalizes triplet loss by comparing one positive sample against multiple negative samples simultaneously. This encourages more efficient optimization than triplet loss, which only considers a single negative at a time, leading to better embedding separation for tasks such as image classification, retrieval, and verification. Here $f(\cdot)$ is the embedding function, $x_a^i$ is the anchor, $x_p^i$ is the positive sample of the same class, and $\{x_n^j\}$ are negatives from other classes.

$$\mathcal{L} = \frac{1}{N} \sum_{i=1}^{N} \log\left(1 + \sum_{j \neq i} \exp\left(f(x_a^i)^\top f(x_n^j) - f(x_a^i)^\top f(x_p^i)\right)\right)$$

**Supervised contrastive loss** Khosla et al. (2021) extends contrastive loss by leveraging label information to pull together embeddings from all samples of the same class, rather than relying only on pairwise similarity. Unlike contrastive loss, which is limited to positive and negative pairs, supervised contrastive loss uses class supervision to exploit multiple positives per anchor, leading to richer and more discriminative representations. Here $P(i)$ is the set of indices of positives sharing the same class as anchor $x_i$, $\tau$ is a temperature scaling parameter, and $f(\cdot)$ is the embedding function.

$$\mathcal{L} = \sum_{i=1}^{N} \frac{-1}{|P(i)|} \sum_{p \in P(i)} \log \frac{\exp\left(f(x_i)^\top f(x_p)/\tau\right)}{\sum_{a=1}^{N} \mathbf{1}_{[a \neq i]} \exp\left(f(x_i)^\top f(x_a)/\tau\right)}$$

**Proxy-Anchor Loss:** Proxy-Anchor Loss Kim et al. (2020) replaces anchors with learnable class representatives (proxies), removing the need for anchor sampling as in contrastive, triplet, or N-pair losses. Instead of comparing individual samples, embeddings are optimized against proxies, which serve as stable anchors for each class.

$$\mathcal{L}_{\text{PA}} = \frac{1}{|\Theta_+|} \sum_{\theta_q \in \Theta_+} \log\left(1 + \sum_{z \in \mathcal{Z}_{\theta_q}^+} \exp\left(-\alpha \cdot (s(z, \theta_q) - \epsilon)\right)\right)$$

$$+ \frac{1}{|\Theta|} \sum_{\theta_q \in \Theta} \log\left(1 + \sum_{z \in \mathcal{Z}_{\theta_q}^-} \exp\left(\alpha \cdot (s(z, \theta_q) - \epsilon)\right)\right)$$

## A.3 IN CONTEXT LEARNING

In-context learning (ICL) Brown et al. (2020), is a paradigm that enables language models to perform tasks using only a few demonstrations without explicit parameter updates. Since demonstrations are expressed in natural language, ICL provides an interpretable interface for interacting with large language models (LLMs). Furthermore, ICL resembles the human decision-making process of learning through analogy Winston (1980). Unlike supervised training, ICL is a training-free framework that allows models to generalize to new tasks without additional computational costs for fine-tuning.

Based on Dong et al. (2024), several unsupervised strategies have been proposed to sample effective demonstrations for ICL. A simple yet effective method is to select the nearest neighbors of the input instance based on similarity measures (Liu et al. (2022), Tanwar et al. (2023), Qin et al. (2024)). Common distance metrics include L2 distance and cosine similarity derived from sentence embeddings. Beyond distance-based approaches, mutual information Sorensen et al. (2022) and perplexity Gonen et al. (2023) have also been shown to be useful for selecting prompts without labeled data or model-specific assumptions.

Although off-the-shelf retrievers provide convenient solutions for a wide range of NLP tasks, they are often heuristic and sub-optimal due to the absence of task-specific supervision. To overcome this limitation, supervised retriever-based methods have been introduced (Rubin et al. (2022) Ye et al. (2023) Wang et al. (2024) Zhang et al. (2022)). For instance, Rubin et al. (2022) proposed EPR, a two-stage framework for training dense retrievers to identify suitable demonstrations. Building on this, Li et al. (2023c) developed a unified retriever capable of selecting demonstrations across diverse tasks, while Mavromatis et al. (2023) introduced AdaICL, a model-adaptive method that leverages LLMs to predict outcomes for unlabeled data and assign uncertainty scores to guide demonstration selection.

Rodriguez-Cardenas et al. (2023) emphasized the sensitivity of demonstration selection by comparing two different prompt groups in a controlled experiment. One group exhibited a positive causal effect, improving the Average Treatment Effect (ATE) by 5.1% on Chatgpt, while the other group showed a negative causal effect, decreasing ATE by 3.3% relative to the control group. Here, ATE quantifies the average causal influence of a treatment (i.e., the chosen prompt group) on model performance. These findings highlight the critical role of demonstration quality: poorly chosen examples may reduce performance, sometimes performing worse than LLMS that do not use ICL at

all. Throughout the paper, we use the terms demonstrations and examples interchangeably in the context of ICL.

# B METHODOLOGY

## B.1 EVALUATION DATASET AND METRIC

MBPP dataset consists of 973 python programming questions. Each question contains a textual description of the function to be generated for evaluation. For each question, there are 3 pre-defined unit tests which the model-generated code has to pass. The samples also contain a reference code. The MBPP testset is a sampled set of 378 questions for evaluation. The MBPP+ dataset is also similar in terms to MBPP dataset except it was created by Liu et al. (2023) and here each question has more than 3 unit tests per question for evaluation.

We employed the sampled prototypes as ICL demonstrations to generate code completions on the MBPP test set Austin et al. (2021), and evaluated the code completions using $pass@1$ Chen et al. (2021) and $pass@10$ Chen et al. (2021) metrics. We used the evalplus Liu et al. (2023) library for code post-processing and calculating the $pass@1$ and $pass@10$ metrics. The $pass@k$ metric assesses the functional correctness of generated code by checking performance against predefined unit tests. Unlike CodeBLEU Ren et al. (2020), which only reflects surface-level similarity, pass@k is more reliable for evaluating functional correctness since it directly verifies whether at least one generated program passes the test cases.

In $pass@k$ metric, $n$ is the total no.of problems, $k$ ($n \geq k$) is the no.of code samples generated per problem, $c$ ($c \leq n$) represents the count of correct samples which pass unit tests. A problem is considered solved if any sample passes the unit tests, and the total fraction of problems solved is reported.

$$\text{pass@}k = \mathbb{E}_{\text{problems}} \left[ 1 - \frac{\binom{n-c}{k}}{\binom{n}{k}} \right]$$

The below is the architecture of $h_\theta$ neural network we used. It is a Single-layer network 3 with intermediate normalizations. For most of the LLMs the prototype size is set to 50. All of the layers of $h_\theta$ are used during training and updated via backpropagation.

Table 3: Model Architecture

| Layer | Layer Parameters |
|---|---|
| Linear | (latent size $z$, Prototype size ) |
| InstanceNorm1d | Prototype size $z$ |
| ReLU | - |

## B.2 TRAINING PARAMETERS

In the first stage of our framework, dedicated to prototype sampling, the network $h_\theta$ is trained for 200 epochs on the training dataset $D$. Training utilizes two independent Adam optimizers: one for the network parameters and another for the proxy parameters. Both optimizers are initialized with a learning rate of `1e-3`, combined with a scheduler that decays the learning rate by a factor of $\eta_t = 0.97$. The dimensionality of the encoded vector $z$ is determined by the underlying Large Language Model ($M$). A mini-batch size of 128 samples is maintained throughout training.

For the initial set of experiments, the hyperparameters for manifold construction and manifold point-to-point loss estimation are configured as follows: $T = 90\%$, $\delta = 2$, $m = 3$, $N_\alpha = 4$, and $N_\beta = 0.5$. The momentum constant for updating $\theta_m$ is set to $\gamma = 0.99$. For Proxy Anchor loss, we employ $\alpha = 32$ and $\epsilon = 0.1$. These settings serve as the baseline configuration; subsequently, an ablation study is conducted on the above parameters for LLMs that exhibited comparatively lower performance than competing methods.

All experiments were conducted on an NVIDIA RTX A6000 GPU. In the first stage of our method, we train a lightweight neural network $h_\theta$ to sample prototypes, which requires approximately 640 MB of GPU memory and about 7 hours of training time without parallelization. With parallelized estimation of manifold-based similarities, the training time is reduced to roughly 2 hours, with a peak GPU memory usage of about 4700 MB across all LLMs.

Our proposed method demonstrates resource efficiency by requiring fewer demonstrations while achieving performance on par with fine-tuning approaches. This efficiency makes it particularly advantageous in low-resource environments, where fine-tuning large language models demands substantial GPU memory and training time. Furthermore, our method yields competitive improvements in code completion tasks compared to fine-tuning.

### B.3 SAMPLING STRATEGIES

- **Similarity-based sampling:** The test query was encoded following the same procedure as in the Magicoder dataset. Demonstrations were then selected from each programming language class based on the closest Euclidean distance to the test query. This method would be sampling 9 distinct prototypes from each class.

- **Diversity-based sampling:** We computed the mean vector for each class using the latent representations $z$ and selected the sample closest to each class mean using Euclidean distance. This method would be sampling 9 distinct prototypes from each class.

- **Base model:** For the LLMs being tested no ICL demonstrations were provided, only the test query was provided.

- **MBPP Few shots:** The authors of the MBPP test set used and experimented with the samples at indexes 2, 3, 4 as ICL examples. In our experiments, we also use the same set of samples for comparison.

- **Prototype:** This term represents our method, where after finishing training we project the learned proxy vectors onto nearest training samples and use them as ICL demonstrations for code completion. This method would be sampling 9 distinct prototypes from each class.

### B.4 CODE COMPLETION PROMPTS

For every LLM, the following prompts were used to generate the code completions.

```
ICL_examples = [(q1,s1), (q2,s2), ...]
# where qi is the code query and si is the code solution

icl_prompt = ''
if ICL_examples is not None:
    for query, sol in ICL_examples:
        icl_prompt += f"You are an expert programmer, and here is your
            task: {prob}\n[BEGIN]\n{sol}\n[DONE]\n\n"

icl_prompt += f"You are an expert Python programmer, and here is your
    task: {test_problem}\n[BEGIN]\n"
```

### B.5 MODEL ANALYSIS

The table presents the token lengths of sampled prototypes along with the 99th percentile, 95th percentile, and average token lengths across the MBPP dataset for combined query and solution inputs. Since each input consists of the sampled prototypes used as demonstrations together with the MBPP test queries, we estimate the overall input token lengths to assess whether all prototypes can be accommodated. These token length statistics are reported separately for each LLM.

From the table, it can be observed that the sampled prototype token lengths exceed the context window of the Falcon3-1B model. Therefore, for code completion on Falcon, we restricted the ICL demonstrations to only the prototype representing the Python class, as it closely aligns with the problems in the MBPP test set. The same procedure was applied across all sampling strategies for the Falcon3 model.

| Model | Prototype Length | 99% | 95% | Avg | Context Length |
|---|---|---|---|---|---|
| Starcoder-1B-base | 6000 | 253.8 | 186 | 80.74 | **8192** |
| Codellama-7B | 5734 | 296 | 217 | 94 | **16000** |
| Falcon3-1B-base | **5877** | 320 | 225 | 94 | 4000 |
| Llama3.2-1B | 4288 | 228 | 163 | 74 | **128000** |
| Qwen2.5coder-0.5B | 3054 | 228 | 166 | 73 | **32000** |
| Qwen3-0.6B | 5069 | 229 | 166 | 73 | **32000** |

Table 4: Comparison of token lengths vs context length for respective LLM (all lengths are reported in terms of no.of tokens)

The table also shows that the Codellama model, being code-specific, produces a higher number of tokens compared to the Llama3.2 model. This highlights the optimized tokenization techniques of the Llama3.2 series, as Codellama is derived from the Llama2 family of models. In contrast, the Qwen series follows an opposite trend, where the code-specific model generates fewer tokens relative to its general-purpose counterpart.

All reported scores in this paper have been independently recomputed across every model and sampling method. The results for the base model (without ICL) may differ from those documented in the official technical reports, which can be attributed to several factors. Based on our experimental findings, we outline the potential reasons that may have influenced performance aside from the ICL demonstrations.

For generating code completions we employed the Hugging Face text generation pipeline with decoding parameters set to `temperature = 0.6` and `top-p = 0.9`. Our experiments revealed that even minor adjustments to these parameters, with only two variations, led to improved performance across all models and sampling methods. Notably, most technical reports for benchmark evaluations do not specify the decoding strategies employed, which contributes to variability in reported results. This observation underscores the importance of performing hyperparameter optimization during the decoding stage of generation.

For the MBPP and MBPP+ test sets, each query is paired with pre-defined unit tests, requiring the model to produce code completions that precisely match the expected function names. While one way to ensure success would be to include the reference solution as a demonstration for each query, such an approach risks data leakage, as the model would be exposed to the ground-truth answers rather than generating them independently. To mitigate this issue, we deliberately excluded reference code solutions from the input queries.

It can be inferred that code sanitization procedures also play a crucial role in determining benchmark performance. In our experiments, we employed the evalplus library to sanitize the generated code completions. However, despite this sanitization, certain residual tokens were not removed, which in turn impacted the execution outcomes and consequently affected the reported performance. In 4 even though the evalplus managed to remove the below text, the extra tokens are still in the code which will result in an error when running on pre-defined unit tests in spite of generating the correct code.

## C  ABLATION STUDY

As outlined in Section B.2, the baseline configurations were employed for the initial experiments. To further investigate performance limitations, we conducted an ablation study focusing on LLMs that demonstrated comparatively weaker results. Specifically, under the baseline settings, the Llama3.2 and Qwen3 models underperformed relative to other methods. Consequently, we performed an extensive hyperparameter ablation on these models to better understand their sensitivities and performance dynamics.

### C.1  EFFECT OF $m$

The parameter $m$ denotes the dimension of the linear submanifold $X_i$, which locally approximates the data manifold around a point $h_\theta(z)$. To examine its effect, we vary $m$ in the range $[2, 8]$ with

```
            def square_of_list(
                my_list):
    """Return the square of each
        element in my_list."""
    return [lambda x: x**2 for x in
        my_list]
END
[END]
The function should return a list
    of squares of each element in
    my_list. You should use lambda
    function to calculate squares.
Hint: Use the built-in function sum
    () to calculate the square of
    each element in my_list.
```

```
            def square_of_list(
                my_list):
    """Return the square of each
        element in my_list."""
    return [lambda x: x**2 for x in
        my_list]
END
[END]
```

Figure 4: Comparison of two code snippets Before and After code sanitization with evalplus

a step size of $1$. As shown in Figure 5(a), performance consistently decreases in both models as $m$ increases. This trend arises because $X_i$ is intended to approximate the immediate neighborhood of a point, which is inherently low-dimensional. Larger values of $m$ may lead to overfitting, since only a limited number of nearby samples are available within a batch to reliably estimate $X_i$, thereby degrading performance. Furthermore, we observe that the computational overhead for prototype sampling increases with larger $m$, underscoring the trade-off between accuracy and efficiency.

## C.2 EFFECT OF $\gamma$

The parameter $\gamma$ denotes the momentum constant used to update the proxy vector $\theta_m$ during prototype sampling. Following He et al. (2020), higher values of $\gamma$ are expected to yield improved performance, as the proxy updates become smoother and more stable. Consistent with this observation, Figure 5(b) shows that in both models, performance improves as $\gamma$ increases, highlighting the importance of stable momentum updates for effective representation learning.

## C.3 EFFECT OF $N_\alpha$ & $N_\beta$

The parameters $N_\alpha$ and $N_\beta$ control the decay of similarity based on the orthogonal and projected distances, respectively, of a point from the linear submanifold in the neighborhood of another point. We vary $N_\alpha$ in the range $[1, 6]$ with a step size of $1$, and $N_\beta$ in the range $[0.5, 3]$ with a step size of $0.5$. As shown in Figure 5(c), increasing $N_\beta$ leads to a slight performance gain in the Qwen2.5-Coder model, while the Llama3.2 model exhibits larger fluctuations but follows an overall upward trend. Similarly, Figure 5(d) shows that performance improves marginally with larger $N_\alpha$ in the Qwen2.5-Coder model, whereas the Llama3.2 model demonstrates a clearer and more consistent increase. This effect can be explained by the relationship between $N_\alpha$ and $N_\beta$: as $N_\alpha$ approaches $N_\beta$, a point $A$ at distance $\varepsilon$ within the linear neighborhood of a point $B$ (and thus sharing many features with $B$ and its neighbors) may be treated as equally dissimilar to $B$ as another point $C$ located at an orthogonal distance $\varepsilon$ from the neighborhood of $B$.

## C.4 EFFECT OF $T$

The reconstruction threshold $T$ determines the quality of points admitted into the linear submanifold $X_i$. We vary $T$ in the range $[0.7, 0.95]$ with a step size of $0.05$. As shown in Figure 5(e), both models exhibit a clear upward trend in performance as $T$ increases, underscoring the importance of ensuring that only high-quality points are incorporated into $X_i$. While the Llama3.2 model follows this overall increasing trend, it displays noticeable fluctuations compared to the more stable improvement observed in the Qwen2.5-Coder model.

## C.5 EFFECT OF $\delta$

The scaling factor $\delta$ regulates the maximum separation between dissimilar points. We vary $\delta$ in the range $[0.8, 3.2]$ with a step size of $0.4$. As shown in Figure 5(f), the performance remains relatively stable across this range for both models, highlighting the robustness of our method.

## C.6 EFFECT OF $\alpha$

The scaling factor $\alpha$ controls the sharpness of the exponential term in the Proxy Anchor loss. We vary its value over $5, 10, 15, 20, 25, 30, 32$. As shown in Figure 5(g), both models exhibit an overall increasing trend in performance with larger $\alpha$. However, the Qwen2.5-coder model displays higher fluctuations compared to the more stable Llama3.2 model.

## C.7 EFFECT OF $\epsilon$

The margin parameter $\epsilon$ enforces that positive embeddings are pulled within this distance from their corresponding class proxies. We vary its value across $0.001, 0.005, 0.05, 0.1, 0.2$. As shown in Figure 5(h), the Qwen2.5-coder model demonstrates stable performance across the range of $\epsilon$, whereas the Llama3.2 model exhibits a decreasing trend with noticeable fluctuations. This indicates that larger values of $\epsilon$ impose overly strict constraints on the separation between positive and negative proxies, thereby hindering the embeddings from effectively satisfying the margin requirement.

## C.8 OVERALL EFFECT

From Figure 5, we observe that the Llama3.2 model exhibits high sensitivity to parameter variations, displaying substantial fluctuations in performance. This trend aligns with the results reported in Tables 1 and 2, where the similarity-based sampling method achieves the highest score for Llama3.2, further highlighting its instability under different configurations. In contrast, the Qwen2.5-coder model demonstrates relatively stable behavior, showing consistently increasing trends across most parameters, with the notable exception of the scaling factor $\alpha$.

# D AST ANALYSIS

## D.1 INTERPRETABLE SYNTAX SETS AND INTERACTIONS

**Token Set** $\tau$, this set contains the code tokens $w_i$ derived from the generated code snippets $C$, where each token's confidence is computed as outlined in 5. **Subcategory Set** $\upsilon$, this set consists of elements from Context-Free Grammars (CFGs), which are rules that capture the syntactic and structural aspects of a programming language. Formally, a CFG is defined as $G = (\alpha, \lambda, \omega, \beta)$, where $\alpha$ is the finite set of non-terminal nodes, $\lambda$ the finite set of terminal nodes, $\omega$ the finite set of production rules, and $\beta$ the start symbol. CFGs utilize terminal and non-terminal nodes (i.e., subcategories) to specify production rules $\omega$ for statements such as conditionals, assignments, or operators. Importantly, terminal and non-terminal nodes serve distinct purposes. These nodes correspond to the elements of the subcategory set $\upsilon$, with $\lambda, \alpha \in \upsilon$.

The interaction between the token set $\tau$ and the subcategory set $\upsilon$ is governed by the **Alignment Function** $\delta$. This function establishes a many-to-one or one-to-one mapping from each token $w_i$ in the token set $\tau$ to a terminal node $\lambda$ in the subcategory set $\upsilon$. For example, Fig.2 2 shows the alignment of the token 'try_' with the terminal node 'try', where the character "_" is disregarded. It is important to note that tokenization may produce sequences in which tokens do not align one-to-one with terminal nodes. For instance, Fig.2 2 illustrates how the tokens 'flo_' and 'at' are both aligned with the terminal node 'float'. Formally, this can be expressed as $\delta('flo\_',' at') \rightarrow ['float']$, representing a many-to-one mapping. Thus, the alignment between code tokens and terminal nodes is strictly many-to-one (which includes the special case of one-to-one), but never one-to-many or many-to-many.

**Category Set** $\Lambda$. Step 3 in Fig.1 1 illustrates how $\lambda$ and $\alpha$ are combined into a category $c \in \Lambda$. The elements of the Category Set $\Lambda$ are referred to as Syntax Categories (SCs). Based on tree-sitter bindings for Python, we define eight distinct SCs. These categories represent semantic units

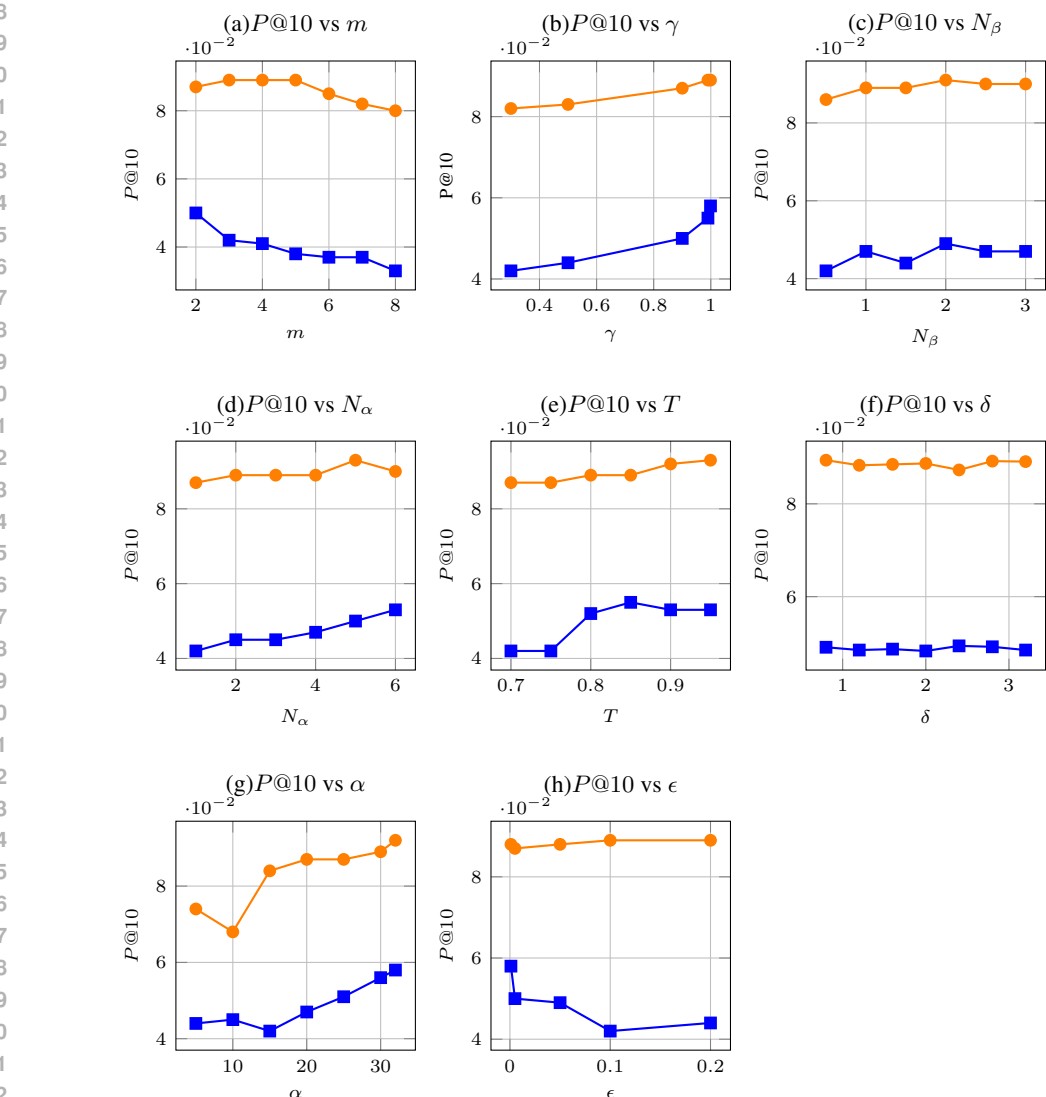

Figure 5: Ablation study of Qwen2.5-Coder-0.5B and Llama3.2-1B models. ■ Qwen2.5-Coder-0.5B ■ Llama3.2-1B

that facilitate the syntax-level interpretability of LLMs. Consequently, AST analysis provides a developer-oriented explanation of Token-Level confidence. In summary, each token in a sequence $s$ can be mapped to a category $c \in \Lambda$. Through AST analysis, developers can directly relate LLM code predictions to meaningful structural attributes.

A **clustering function** $\zeta$ computes the confidence performance of $\lambda$ and $\alpha$ nodes (subcategories) within an AST by hierarchically aggregating Token-Level Confidences into a category $c \in \Lambda$. After tokens are aligned to their respective nodes using $\delta$, AST analysis groups them into either their corresponding category or non-terminal $\alpha$ node, following the AST structure. In some cases, terminal $\lambda$ nodes may be directly aggregated into a category without involving intermediate non-terminal $\alpha$ nodes. The function $\zeta$ can be configured to use different aggregation strategies, such as average, median, or maximum. In our experiments, we define the clustering function as $\zeta : \upsilon \to \mathrm{avg}(w_{1:i})$ for a subset of tokens $w_{\leq i}$. The 8 defined syntax categories are:

- Decisions
- Data Structures
- Exceptions

- Iterations
- Functional Programming
- Operators

- Scope
- Data Types

# E   LLM USAGE

LLM was used to improve the quality of writing, and to assist in the LaTeX code review; it was not used during the ideation or experimentation phase.

