# OpenReview forum: "Protocode: Prototype-Driven Interpretability for Code Generation in LLMs"
_ICLR.cc/2026/Conference — ICLR 2026 Conference Withdrawn Submission_

### Official Review · Reviewer_pk5F · 2025-10-27

**Soundness:** 2
**Presentation:** 1
**Contribution:** 2
**Rating:** 2
**Confidence:** 3

**Summary:**

This paper explores automatic selection of in-context learning (ICL) examples for code generation.
The method analyzes AST structures to identify influential code regions, performs clustering based on these structural cues, and uses representative prototypes to construct ICL contexts.

**Strengths:**

- Addresses a practical and timely problem—automatically identifying effective ICL examples for code generation.

- Leveraging abstract syntax trees provides a structured, semantics-aware notion of code similarity.

**Weaknesses:**

- Methodological clarity: The procedure for prototype selection—an important step within the overall clustering framework—is described indirectly and spread across several sections (e.g., L145–147, L997–999).
Consolidating these details into one cohesive subsection or including pseudo-code would make the approach clearer and easier to reproduce.

- Evaluation scope:
While prototypes are defined per programming language, experiments focus solely on Python.
Clarifying whether the clustering was performed cross-language or language-specific would help interpret generalizability.

- Overall performance:
The reported Codellama-7B Pass@1 (≈ 2.4–3.0) is much lower than commonly reported results on MBPP / MBPP+ (≈ 47–57) [1].
This substantial gap suggests that the experimental setup or evaluation procedure may need to be revisited.

- Static ICL selection:
The method appears to rely on a fixed set of in-context examples independent of the input query, which limits adaptivity and may understate performance.

- Representation choice:
Decoder-only hidden states are optimized for next-token prediction rather than holistic semantics, which can limit clustering or retrieval quality when reused as embeddings [2].

[1] https://evalplus.github.io/leaderboard.html

[2] Parishad BehnamGhader, Vaibhav Adlakha, Marius Mosbach, Dzmitry Bahdanau, Nicolas Chapados, & Siva Reddy (2024). LLM2Vec: Large Language Models Are Secretly Powerful Text Encoders. In First Conference on Language Modeling.

**Questions:**

- Please provide a concise diagram summarizing clustering and prototype retrieval steps.

- Check the evaluation setup with standard MBPP / MBPP+ protocols.

- Extend experiments (or discussion) to cross-language settings to validate broader applicability.

- Compare static vs. query-adaptive selection to see how dynamic retrieval might help.

- Motivate embedding choices and their implications for clustering quality.

- Substantiate interpretability claims with qualitative or human-study examples illustrating AST-based influence.

---

### Official Review · Reviewer_5nLH · 2025-10-30

**Soundness:** 3
**Presentation:** 1
**Contribution:** 2
**Rating:** 4
**Confidence:** 4

**Summary:**

The authors present ProtoCode a method for automatically sampling In-Context Learning demonstrations in order to improve the performance of coding models, as well as the interpretability of generations. A novel AST-based analysis identifies regions of code most influenced by the in context samples. The method combines piecewise-linear manifold learning with proxy-anchor metric learning to automatically sample high-quality ICL samples. This approach produces ICL samples that capture local data structures and are semantically meaningful. Additionally, for interpretability they map prototype-gradient attribution to ASTs, which avoids storing the full token probability distribution. In the experiments, the authors evaluate their approach on the MBPP dataset using 6 different LLMs, showing modest improvements in pass@10 metrics while enabling syntax-grounded explanations of which code regions are most influenced by the chosen demonstrations.

**Strengths:**

- The related work and review of prior work is well written and clearly places the current methods within literature.
- The evalaution setup spans 6 models, including both general-purpose and code-specific models. The experimental setup supports a reasonable generalizability of their approach. Likewise, the baselines include no ICL, similarity-based samples, diversity-based samples, MBPP provides samples which gives a good comparison. However, I would like to see these baseline methods introduced in the main section of the paper with, perhaps, more details on them left to Appendix.
- The ablation study from the Figures in Section C demonstrate sensitivity analysis across 8 hyperparameters.
- Experimental confounders (e.g., santization) and mentioning limitations, such as those discussed in `B.5`, are important to detail for reader and good to see.
- The care take around reproducibility is great — the appendices include extensive implementation details.

**Weaknesses:**

- The presentation could be improved, specifically a figure to help readers grasp the high-level approach and novelty of the method.
- Additionally, it's difficult to grasp the core message from the current figures. Some additional context, especially in results figures like Figure 3 and Table 1 to clearly state what is being shown and why it is significant, and the axes must be labeled.
- The performance gains seem modest (but consistent! Prototypes is best or 2nd best for all models), however it may just be that the gains are difficult to assess since none of the models perform super well on MBPP.
- The experiments lack any statistical significance. Table 1 doesn't include confidence intervals, and Figure 3 lacks error bars and is difficult to compare the results. Given that the performances for all models are low, do the results mean very small differences in correctness (e.g., only a handful of samples as being correct/incorrect between the different models)?
- For the Interpretability task it's difficult to assess the signficance without some level of human evaluation or major qualitative evaluation of the results (or how it might compare).

**Questions:**

1. Can you clarify why piecewise-linear manifold learning is necessary. Would simpler methods like PCA suffice?
2. Do you have any intuitions for why similarity-based sampling outperforms your method for Llama3.2?

---

### Official Review · Reviewer_PC6h · 2025-10-31

**Soundness:** 2
**Presentation:** 2
**Contribution:** 2
**Rating:** 2
**Confidence:** 3

**Summary:**

Motivated by the lack of faithful interpretability in existing methods, the authors propose ProtoCode, which is a prototype-based in-context learning (ICL) sampling approach grounded in manifold learning and metric learning. Their method automatically selects representative demonstrations (“prototypes”) that are geometrically consistent and semantically discriminative. Using gradient-based attribution between prototype and token embeddings, the approach maps influence scores to Abstract Syntax Tree (AST) nodes, yielding syntax-grounded explanations at both local (node-level) and global (category-level) scales. Experiments across six models on MBPP and MBPP+ benchmarks show that prototype sampling consistently improves pass@10 accuracy compared to other baselines. The AST analysis further reveals consistent confidence alignment between prototype influence and key syntax categories (e.g., functions, scope, data structures). The study concludes that prototype-driven interpretability provides both efficiency and explanatory power for code generation.

**Strengths:**

1. Unlike previous work focusing solely on accuracy, the proposed method is designed to improve both the interpretability of model behavior and the quality of code generation by selecting meaningful in-context examples.
2. The paper integrates piecewise-linear manifold learning with proxy-anchor metric learning in a unified framework, allowing prototypes to capture both geometric structure and semantic distinctions within the data.
3. The paper introduces a prototype–token gradient attribution mechanism that links each prototype to specific parts of the generated code. This allows the model to produce syntax-aware and transparent explanations of its behavior.
4. The AST-based evaluation examines interpretability across syntax categories such as decisions, iterations, and exception handling. It gives a detailed view of how different components of a program are influenced by the model.
5. The approach is grounded in the manifold hypothesis and metric learning theory. It provides a clear and mathematically supported explanation of how prototypes are formed and applied.

**Weaknesses:**

1. The paper lists four baseline sampling strategies — base, diversity, similarity, and mbpp — in Tables 1 and 2, but provides no explanation of how these baselines are implemented or what criteria they use to select in-context examples.
2. The baseline performance reported in the paper are much lower than the official performance numbers for the same models. This difference suggests possible issues in the evaluation setup. For example, the paper shows Qwen3-0.6B reaching only 1.1% pass@1 and 4.8% pass@10 on MBPP (Table 1) under the “base” setting. In contrast, the official Qwen-2.5 technical report (page 8, Table 8, https://arxiv.org/pdf/2505.09388) reports 36.6% for the Qwen3-0.6B model under a 3-shot MBPP setting. The same issue appears for CodeLlama-7B, which achieves only 2.1% pass@1 and 11.6% pass@10 under the “base” setting in this paper, while the CodeLlama report (page 6, Table 2, https://arxiv.org/pdf/2308.12950) shows 41.4% pass@1 and 61.7% pass@10. These large differences suggest that the evaluation in this paper does not align with standard MBPP protocols and may undervalue the true baseline capability of the models.
3. While the paper notes that poor prototypes can degrade performance, it does not include qualitative examples showing when or why the prototype selection fails. This omission weakens understanding of the method’s limitations
4. Although syntax categories are analyzed, the paper never discusses whether higher syntax confidence correlates with better code correctness, leaving the relationship between interpretability and performance unverified

**Questions:**

1. A major concern lies in the evaluation results presented in Section 4. Some implementation details appear missing, which makes it difficult to interpret the reported numbers. Could the authors clarify how the evaluation was conducted and how each baseline was implemented? How those number were calculated?

---

### Official Review · Reviewer_i6KR · 2025-11-02

**Soundness:** 3
**Presentation:** 2
**Contribution:** 2
**Rating:** 2
**Confidence:** 3

**Summary:**

This paper introduces PROTOCODE, a prototype-driven framework that enhances interpretability in large language models (LLMs) for code generation. Specifically, the method can automatically sample in-context learning (ICL) demonstrations through a joint manifold–metric learning framework. Moreover, the method combines it with AST-grounded prototype attribution to identify syntactic regions of generated code most influenced by specific prototypes.

**Strengths:**

1. The idea is interesting, compared to previous works, which focused on either attribution or syntactic structure, this work connects them via prototype influence scores.
2. The author provides extensive experiments, covering 6 LLMs and two datasets. Moreover, the AST-based interpretability analysis is detailed and interpretable, showing how models differ in their syntactic confidence across categories like functions, iteration, and exception handling

**Weaknesses:**

1. The evaluation metrics for the paper are not sufficient. While the author emphasizes the method's interpretability, the main evaluation focuses on syntactic confidence distributions rather than on whether the attributions faithfully reflect the model’s internal causal mechanisms. Specifically, the author didn't provide sufficient evidence to support the experiment's results. To provide stronger evidence of the methods' interpretability, the author should include additional ablation studies. For example, in section 5.2, the author shows that Qwen exhibits highly consistent confidence across all syntax categories, then the author should test if we remove these categories from the prototypes and regenerate the same code and see whether the results change a lot.
2. It is better to provide more intuitive explanations to describe why the architecture is designed. For example, in section 3.4, the author provides only the loss function without explaining why it is designed this way.
3. Some small weakness: the figure can be further improved, like figure 2, some words are out of the box. For Figure 3, I don't understand why the author split the 6 LLMs into 3 subfigures rather than drawing 1 figure that would help the reader better compare them.

**Questions:**

If the author can provide more ablation studies to prove its $\textbf{interpretability}$, I would like to raise my score. Current results cannot fully demonstrate the methods' contribution.

---

### Note · Authors · 2025-12-03

I have read and agree with the venue's withdrawal policy on behalf of myself and my co-authors.